# Polysaccharide-Stabilized PdAg Nanocatalysts for Hydrogenation of 2-Hexyn-1-ol

Alima K. Zharmagambetova [ID], Eldar T. Talgatov [ID], Assemgul S. Auyezkhanova *[ID], Farida U. Bukharbayeva [ID] and Aigul I. Jumekeyeva [ID]

Laboratory of Organic Catalysis, D.V. Sokolsky Institute of Fuel, Catalysis, and Electrochemistry, Almaty 050010, Kazakhstan; a.zharmagambetova@ifce.kz (A.K.Z.); e.talgatov@ifce.kz (E.T.T.); farida.isakova1992@gmail.com (F.U.B.); jumekeeva@mail.ru (A.I.J.)
* Correspondence: a.assemgul@mail.ru

**Abstract:** We used a new one-pot green technique to prepare polysaccharide-based Pd and PdAg nanocatalysts. Catalysts were obtained using a sequentially supporting natural polymer (2-hydroxyethyl cellulose (HEC), chitosan (Chit), pectin (Pec)) and metals on zinc oxide. Nanocatalysts based on a polysaccharide were studied using various physicochemical methods (IR spectroscopy, transmission electron microscopy, X-ray powder diffraction, etc.). The catalyst characterization results indicated the complete adsorption of polysaccharides and metal ions onto the inorganic support (ZnO). We demonstrated the formation of polysaccharide-stabilized Pd nanoparticles with a size of ~2 nm. Metal nanoparticles were uniformly located on the surface of polysaccharide-modified zinc oxide. The synthesized catalysts were tested using liquid-phase hydrogenation of 2-hexyn-1-ol under mild conditions (0.1 MPa, 40 °C). Close conversion values of 2-hexyn1-ol were obtained for all the developed catalysts. The selectivity for *cis*-hexen-1-ol of the polysaccharide-based PdAg nanocatalysts varied as follows: PdAg-HEC/ZnO > PdAg-Pec/ZnO > PdAg-Chit/ZnO. A similar correlation was obtained in the presence of monometallic Pd-polysaccharide/ZnO catalysts. We determined the optimum reaction temperature and catalyst loading for PdAg catalysts modified using HEC and Chit (40 °C, 0.05 g).

**Keywords:** hydrogenation; palladium–silver catalysts; polysaccharide; 2-hexyn-1-ol

## 1. Introduction

Because of the requirements of green chemistry, researchers in the field are currently focused on the study of nontoxic natural polymers to create new environmentally friendly nanomaterials and stabilized metal nanoparticles for widespread use [1–9]. Polysaccharides derived from natural sources are suitable alternatives to synthetic polymers produced from petroleum products [5–9]. Polysaccharides have different functional groups in their structure and can, therefore, form composites with mineral sorbents [10–15] and transition-metal ions [15–18].

For example, the hydroxyl and amino groups in the chitosan structure, interacting with transition-metal ions, form metal nanoparticles, which are very promising as catalysts [19,20]. The multifunctionality of pectin (Pec) is due to the nature of its molecule, which consists of linear chains of 1,4-linked residues of $\alpha$-d-galacturonic acid [3,21–23] with a large number of $-OH^-$ and $-COOH^-$ groups. This nature allows its use in the design of various polymer–inorganic materials such as nanocatalysts [3], sorbents for wastewater treatment [4], thickeners, emulsifiers, and gelling agents for the food industry and biomedicine [21–23].

Cellulose is the most common natural polysaccharide. Its soluble derivative 2-hydroxyethyl cellulose (HEC) is formed by treating cellulose with alkali and causing it to react with ethylene oxide [24]. The advantages of this derivative are its good water solubility, biodegradability, biocompatibility, and film formation [25–28]. It is widely used in pharmaceuticals [29], the textile industry [30], paper making [31], cosmetology [32], etc.

The use of polysaccharides as new auxiliary materials for the design of heterogeneous catalysts is becoming more frequent. Chitosan (Chit) is beginning to find use [8], primarily due to its high affinity for metal ions [20]. Several studies have emerged that are devoted to catalysts for processes such as hydrogenation [1,33,34], oxidation [3,4], coupling reactions, and other possible catalytic syntheses [8,19,20,35]. Chitosan acts as a reducing agent in catalysts due to the presence of functional groups ($CH_2OH$ and $NH_2$) in its structure [36]. Chitosan can also stabilize catalytic centers and prevent the formation of aggregates and the occurrence of metal leaching [33,36,37]. In Ref. [33], an efficient chitosan-based Ni catalyst for the selective hydrogenation of acetylene was developed. The Ni catalyst was obtained through the interaction of a chitosan/carbon nanotube composite with a $NiSO_4$ solution. Using the Ni-chitosan/carbon nanotube catalyst, the acetylene conversion and ethylene selectivity reached 100% at temperatures of 160 and 190 °C, respectively. In Ref. [38], unmodified commercial chitosan was investigated as a heterogeneous catalyst for the transfer reaction of diazo compounds. Various compounds (1,3-diketones, malonates and β-ketoesters) were selected as substrates. The catalyst showed stability and could be easily recovered via filtration after a simple treatment. In Ref. [1], chitosan was used as a support to create palladium nanocatalysts. The palladium particles were uniformly distributed across chitosan. The chitosan-based catalysts (Pd@CS) were investigated in the hydrogenation of 4-nitrophenol and compared with commercial catalysts (Pd/C). The synthesized chitosan-based palladium catalysts showed high catalytic activity and stability in the selective hydrogenation of aromatic aldehydes. In Ref. [39], chitosan was modified by mixing it with polyvinyl alcohol. We prepared palladium catalysts based on chitosan–polyvinyl alcohol using a reduction method. Subsequently, they were utilized for the Suzuki conjugation reaction of halogenated benzene and phenylboronic acid. The catalyst exhibited good substrate stability and catalytic activity. One of the disadvantages of chitosan-based catalysts is the low strength of chitosan microspheres, which can be easily overcome during the reaction. Ref. [40] proposed an environmentally friendly method for the preparation of a heterogeneous nanocatalyst based on palladium nanoparticles anchored on chitosan/$Co_3O_4$ (CS/$Co_3O_4$). The developed PdNPs/CS/$Co_3O_4$ was successfully applied in the cyanidation of aryl halides to various substituted benzonitriles.

There are limited data on pectin-containing catalysts or their design and use as stabilizers for metal nanoparticles [2–4,41–44]. In Ref. [45], an environmentally friendly method for the production of pectin-based palladium nanoparticles was developed. Palladium nanoparticles, stabilized using pectin (Pdnp/pectin), were prepared by exposing an aqueous solution of $PdCl_2$ (100 mL, 1 mM) to pectin without an additional reducing agent. The synthesized nanoparticles were investigated in the Mizoroki-Heck reaction between various aryl halides and n-butyl acrylate under solvent-free conditions. The catalyst can be reused for six cycles without significant loss of catalytic activity. In Ref. [46], pectin was used as a support for the preparation of a heterogeneous catalyst based on metal phthalocyanines. Before the immobilization of copper tetraaminophthalocyanine, pectin was oxidized using periodate. The catalytic activity of the developed catalysts was investigated in a $CO_2$ fixation reaction to produce cyclic carbonates. The cation-binding ability of pectin was investigated in Ref. [47]. A CaO-based catalyst was prepared in accordance with the precipitation method using $Na_2CO_3$ and $Ca(NO_3)_2$ in the presence of pectin, which was followed by calcination. The most active catalyst, CaP-600, was prepared at 600 °C, and it showed a high level of activity in the transesterification reaction of soybean oil to produce biodiesel.

Different types of cellulose have been used to support catalysts in hydrogenation, oxidation, dye reduction, and coupling reactions [3,4,35,48–51]. Bearing abundant reactive –OH groups on its chains, HEC can act as both a reducing agent for transition-metal ions and a stabilizing agent for the metal nanoparticles formed. Despite the fact that HEC is an attractive biopolymer [26–28], its full potential for use in the design of heterogeneous catalysts has not yet been adequately explored [52,53]. Thus, [24] reports on the synthesis of an HEC-modified palladium catalyst, which was successfully tested via Suzuki reactions

in aqueous ethanol solution. In [53], ruthenium (Ru) nanoparticles synthesized from hydroxyethylcellulose (HEC-Ru) were used in the hydrogenation of a-pinene in aqueous medium. The results showed that HEC promoted the dispersion of Ru nanoparticles, with the particle sizes ranging from 4 to 6 nm. The micelles formed with HEC acted as "microreactors". In addition, a-pinene was loaded into the HEC micelles through hydrophobic interaction, resulting in promising contact with Ru nanoparticles. Thus, the HEC-Ru nanoparticles significantly improved the hydrogenation of a-pinene, with a-pinene conversion and *cis*-pinane selectivity reaching 99.6% and 98.6%, respectively.

Thus, there exists a sufficient number of works devoted to the development of polysaccharide-modified catalysts for various catalytic processes. However, not much has been reported on the application of such catalysts in the hydrogenation of acetylene alcohols. The selective hydrogenation of alkynol through carbon-carbon triple bonds is an important process involved in producing fine chemicals and pharmaceuticals [54,55].

Among metal catalysts, supported Pd is commonly applied in this process due to its high activity. However, it has the disadvantage of insufficient selectivity for alkenols at high conversions. The addition of metal (Ag, Au, Cu or Zn) can improve Pd catalysts due to the formation of small palladium nanoparticles [56–58]. Another approach for improving the effectiveness of Pd catalysts is the use of functional polymers to stabilize and modify active centers [59,60].

In order to assess the growing need for environmentally friendly nanocatalyst syntheses, this work aimed to prepare polysaccharide-containing PdAg/ZnO catalysts. HEC, pectin and chitosan were used as green stabilizers, and water was the only medium for catalyst synthesis in ambient conditions without high-temperature processes of calcination and reduction. To this end, we used a new, green one-pot technique for catalyst synthesis via sequentially supporting polysaccharides and metals on zinc oxide. In this study, we evaluated and compared the efficiency of the developed HEC-, Chit- and Pec-stabilized bimetallic PdAg catalysts supported on zinc oxide with that of the monometallic Pd-polysaccharide/ZnO nanocatalysts in the hydrogenation of 2-hexyn-1-ol.

## 2. Results

### 2.1. Characterization of Catalysts

Mono- and bimetallic palladium and palladium–silver catalysts, modified using polysaccharides such as chitosan (Chit), 2-hydroxyethyl cellulose (HEC) and pectin (Pec), were prepared via adsorption methods in an aqueous medium under ambient conditions and constant stirring. The solution of polymers and metal salts was sequentially added into a zinc oxide suspension. The resulting composites were washed with water and dried in air. As a result, the following catalysts were obtained: Pd-HEC/ZnO, Pd-Chit/ZnO, Pd-Pec/ZnO, PdAg-HEC/ZnO, PdAg-Chit/ZnO and PdAg-Pec/ZnO.

The palladium and silver content in the catalysts were evaluated using spectrophotometry and potentiometry methods, respectively. Analyses of the supernatant solution before and after the sorption process showed that 91–99% of the introduced Pd and 99–100% of Ag were adsorbed onto the polymer-modified ZnO. The calculated total metal content (Pd and Ag) in the obtained polymer-modified mono- and bimetallic catalysts was 0.46–0.49 wt%, which is close to the expected value of 0.5 wt% (Table S1).

Table 1 shows the results of an elemental analysis of the catalysts. The palladium contents in Pd-HEC/ZnO, Pd-Chit/ZnO, Pd-Pec/ZnO, PdAg-HEC/ZnO, PdAg-Chit/ZnO and PdAg-Pec/ZnO catalysts were found to be 0.49%, 0.47%, 0.57%, 0.36%, 0.44% and 0.48%, respectively. The silver contents in the bimetallic PdAg-HEC/ZnO, PdAg-Chit/ZnO and PdAg-Pec/ZnO catalysts were found to be 0.18%, 0.15% and 0.16 %, respectively. Thus, the total metal content in all catalysts was no lower than 0.5 wt%, suggesting the almost complete adsorption of metals (Pd and Ag) onto the polymer-modified support materials (Table 1). This is consistent with the data obtained using spectrophotometry and potentiometry methods.

**Table 1.** Results of elemental analysis of the catalysts obtained.

| Sample | Elemental Composition of the Catalyst, wt% | | |
|---|---|---|---|
| | Pd$_{calcd/detd}$ | Ag$_{calcd/detd}$ | Zn$_{calcd/detd}$ |
| Pd-HEC/ZnO | 0.50/0.49 | - | 79.0/77.8 |
| Pd-Chit/ZnO | 0.50/0.47 | - | 79.0/81.1 |
| Pd-Pec/ZnO | 0.50/0.57 | - | 79.0/80.5 |
| PdAg-HEC/ZnO | 0.37/0.36 | 0.13/0.18 | 79.0/81.5 |
| PdAg-Chit/ZnO | 0.37/0.44 | 0.13/0.15 | 79.0/81.6 |
| PdAg-Pec/ZnO | 0.37/0.48 | 0.13/0.16 | 79.0/82.0 |

The results of X-ray powder diffraction analysis (XRD) of the ZnO, HEC/ZnO and PdAg-HEC/ZnO are shown in Figure 1. All XRD patterns showed characteristic peaks at 37.0°, 40.2°, 42.3°, 55.8°, 66.7°, 74.5°, 78.9°, 80.9°, 82.3°, 86.8° and 92.6°, corresponding to the (100), (002), (101), (102), (110), (103), (200), (112), (201), (004) and (202) planes of ZnO wurtzite structures (JCPDS card no. 79-0206) [61]. The broad peak observed at 23° in the polymer-modified materials could be attributed to the amorphous phase of the HEC [62]. No peaks related to palladium (Pd or PdO) and silver (Ag or AgCl) species were observed on the XRD pattern of the PdAg-HEC/ZnO catalyst. This can be explained by the low metal (Pd and Ag) content in the catalyst and the small particle sizes [63].

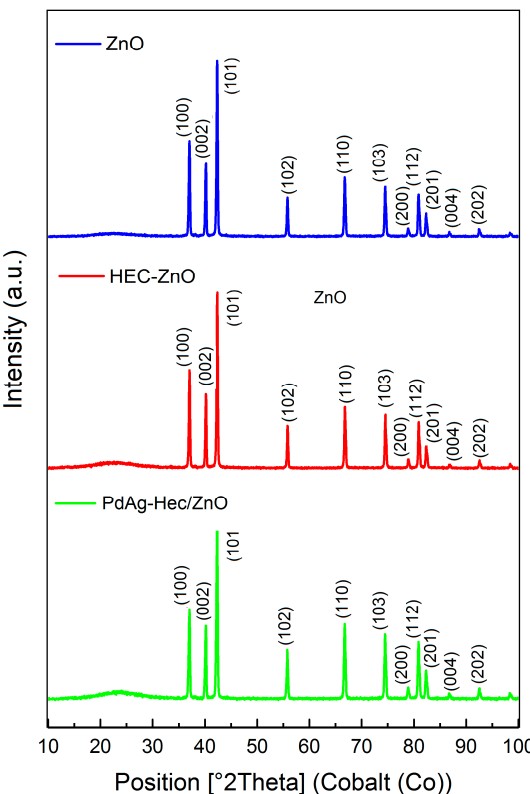

**Figure 1.** XRD of ZnO, HEC/ZnO and PdAg-HEC/ZnO.

The modification of ZnO using HEC was also confirmed via the standardized Brunauer-Emmett-Teller (BET) method (Table 2). HEC/ZnO and PdAg-HEC/ZnO composites were characterized by decreased surface area in comparison with the initial zinc oxide. It should be noted that the PdAg-HEC/ZnO catalyst demonstrated a higher surface area compared with that of the HEC/ZnO composite. This is consistent with data obtained for similar systems modified with polyvinylpirollidone (PVP) [64]. In Ref. [64], the decrease in the surface area from ZnO to PVP/ZnO was attributed to the blockage of micropores in inorganic material after modification with the polymer. Conversely, the increase in the surface area

from PVP/ZnO to Pd-PVP/ZnO could be attributed to a decrease in the surface coverage of the ZnO with a PVP shell via alterations in the orientation of the polymer functional groups from ZnO to Pd. Another potential explanation for such changes in the surface area from unmodified zinc oxide to a polymer-modified catalyst was related to the changing degree of agglomeration of ZnO particles after modification with a polymer and following the adsorption of metal ions onto the polymer/ZnO composite. That is, adding an HEC solution to a zinc oxide suspension can lead to the agglomeration of ZnO particles, which probably occurs due to ZnO-HEC-ZnO bonding. The adsorption of metal ions onto the HEC/ZnO composite, on the contrary, can decrease the agglomeration of ZnO particles due to the formation of ZnO-HEC-Pd (Ag) bonds. It should be noted that according to these assumptions, HEC can interact with both zinc oxide and metal (Pd and Ag) ions.

**Table 2.** Surface area of ZnO, HEC/ZnO and PdAg-HEC/ZnO.

| Sample | Surface Area, $m^2\ g^{-1}$ |
| --- | --- |
| ZnO | 8.7 |
| HEC/ZnO | 1.1 |
| PdAg-HEC/ZnO | 5.2 |
| Pd-HEC/ZnO | 4.9 |

　　　The interaction of the polysaccharide with other components of the PdAg-HEC/ZnO catalyst and the formation of a polymer—metal complex on the ZnO surface was confirmed using infrared spectroscopy (IRS). Figure 2 shows the IR spectra of HEC, HEC/ZnO and PdAg-HEC/ZnO. HEC shows characteristic bands at 1061 and 1122 $cm^{-1}$, corresponding to the C–O–C stretching vibrations in the glucopyranose structure and C–O anti-symmetric vibrations, respectively [65,66].

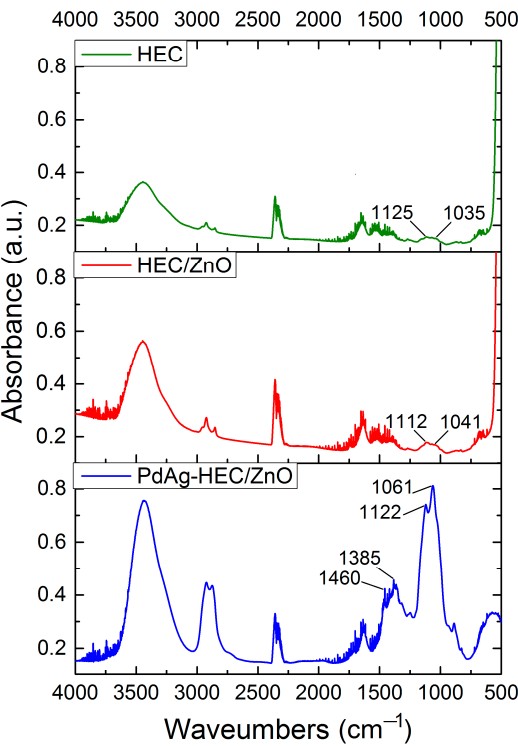

**Figure 2.** IR spectra of the HEC, HEC/ZnO and PdAg-HEC/ZnO.

　　　Other characteristic bands at 1460 and 1385 $cm^{-1}$ were attributed to O–H "plane deformation" and C–H "symmetric bending vibrations", respectively, in –$CH_2O$– [65,66]. The shift in the absorption bands of the O–H, C–O–C, C–O and –$CH_2O$– groups in the IR spectrum of the catalyst, compared to the same bands in HEC and HEC/ZnO (Figure 2,

spectra 1 to 3), confirmed the intermolecular interaction of HEC with both Pd (or Ag) ions and $Zn^{2+}$ of ZnO through van der Waals forces.

In our prior studies [64,67], similar 1%Pd-PVP/ZnO, 1%Pd-PAM/ZnO, 1%Pd-Ag (3:1)-PAM/ZnO and 1%Pd-Ag (3:1)-HEC/ZnO catalysts were characterized via the TEM method. We showed that the interaction of polymers with metal ions on a support surface led to the formation of smaller active-phase nanoparticles (~2 nm) compared with similar unmodified supported catalysts. The transmission electron microscopy (TEM) image of the Pd-HEC/ZnO catalyst showed the formation of finely dispersed palladium nanoparticles of ~2 nm, which were evenly distributed on the surface of zinc oxide modified with HEC (Figure 3a). This was also confirmed using SEM and EDX elemental mapping images of Zn and Pd from the Pd-HEC/ZnO catalyst (Figure 3b), according to which palladium and zinc were homogeneously distributed. It should be noted that reflections corresponding to $PdO_xH_2O$ (JCPDS 9-254), $PdO_2$ (JCPDS 34-1101), PdZn (JCPDS 6-620), $PdZn_2$ (JCPDS 31-942), $PdZn_2O_4$ (JCPDS 32-723), Ag (JCPDS 4-783), $Ag_2O$ (JCPDS 19-1155), AgZn (JCPDS 29-1156) and $AgZn_3$ (JCPDS 25-1325) phases were observed in the diffraction patterns obtained from the TEM images of Pd-HEC/ZnO and PdAg-HEC/ZnO catalysts (Tables S1 and S2). Thus, the electron microscopy results show that metal ions can be stabilized via the polysaccharide, in addition to interacting with ZnO in the formation of new phases.

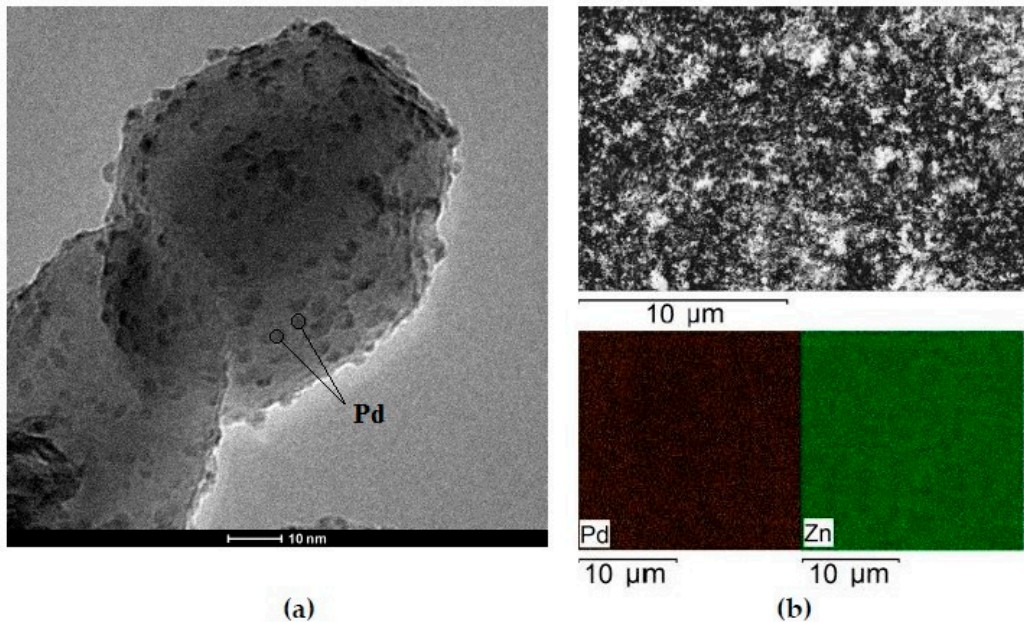

**Figure 3.** TEM microphotograph (**a**) and SEM/EDX elemental mapping images (**b**) of Pd-HEC/ZnO catalyst.

Studying the Pd-HEC/ZnO catalyst using X-ray photoelectron spectroscopy (XPS) also confirmed the presence of both polysaccharides and palladium on the surface of zinc oxide (Figure 4).

An analysis of the XPS spectrum of the Pd 3d (5/2) region (Figure 4a) showed the presence of palladium in the catalyst o in both oxidized ($Pd^{2+}$) and reduced ($Pd^0$) states at binding energies of ~337.3 and ~336.0 eV, respectively [68]. The presence of a small amount of zero-valent palladium was probably caused via the photocatalytic reduction of $Pd^{2+}$ in the presence of zinc oxide, which is known as an efficient photocatalyst [69]. This suggested that a small part of the introduced palladium interacted with zinc oxide. This was confirmed by the fact that the binding energy of $Pd^0$ had a positive shift (ca. 0.6 eV) due to a strong metal–support interaction and the formation of PdZn species [70], which was consistent with the microdiffraction analysis data (Tables S2 and S3). In the case of palladium in an oxidized state, such shifting was not observed, confirming the stabilization of $Pd^0$ particles

using the polymer. The peaks at 1020.4 and 1045.0 eV (Figure 4b), corresponding to Zn 2p (3/2) and Zn 2p (1/2), respectively, were attributed to the presence of $Zn^{2+}$ ions in a zinc oxide crystal lattice [71]. The peaks shifted toward smaller energies in comparison with typical ZnO [72,73], a phenomenon attributed to the interaction of $Zn^{2+}$ with the oxygen-containing functional groups of HEC. It should be noted that similar negative shifting was observed in nanosized zinc oxide particles [74,75].

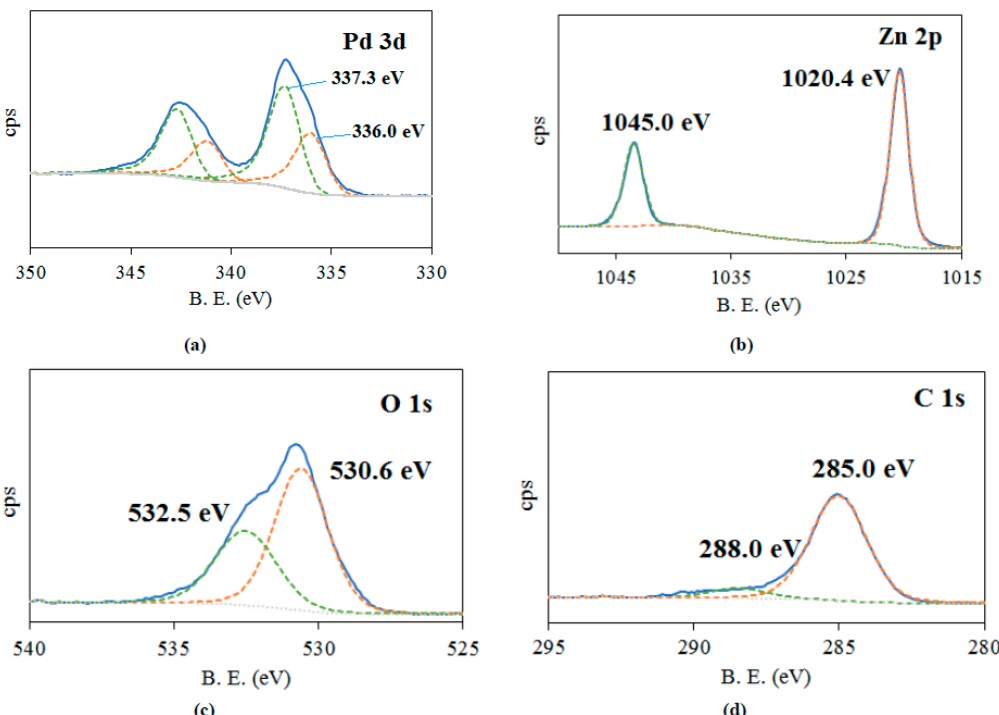

**Figure 4.** The lines of the Pd 3d (5/2) (**a**), Zn 2p (3/2) (**b**), O 1s (**c**) and C 1s (**d**) from the XPS spectrum of the Pd-HEC/ZnO catalyst. Blue curve is main analytical line. Green dashed curve is 532.5 eV line. Brown dashed curve is 530.6 eV line.

The XPS spectrum of the O 1 s region (Figure 4c) was de-convoluted into two peaks with binding energies of 530.6 (peak 1) and 532.5 eV (peak 2). According to the literature data [72], peak 1 is related to $O^{2-}$ ions in the Zn–O bonding of the wurtzite structure, and peak 2 is attributable to the presence of oxygen from OH groups on the zinc oxide surface. Elsewhere, peak 2 was assigned to the C–O oxygen in the HEC repeated unit [73]. The de-convolution of the C 1s line in the XPS spectrum of the catalyst indicated that carbon was represented by two components (Figure 4d). The main component centered at 285.0 eV was attributed to the C–C bond in the HEC macromolecule, and no shift in the binding energy was observed for this peak. In the case of the minor peak at 288.0 eV related to the C–O bond in HEC, some positive shift in binding energy (ca. 1.3 eV) was observed, which is probably due to the interaction of the C–O–C, C–O and –$CH_2O$– groups of HEC with both $Pd^{2+}$ and $Zn^{2+}$ of zinc oxide [73].

Thus, the characterization of polysaccharide-modified catalysts using physical–chemical methods such as spectrophotometry, potentiometry, elemental analysis, XRD, IRS, BET, TEM and XPS indicated that polysaccharide and metal (Pd and Ag) ions were quantitatively adsorbed onto zinc oxide and that the polymers interacted with both ZnO and the active-phase particles formed. The role of polymers was both to fix Pd and Ag species onto the zinc oxide surface and guarantee their stabilization. This was consistent with the data obtained in our prior study for similar Pd-PVP/ZnO catalysts [64]. However, in the case of Pd-HEC/ZnO, a small amount of Pd also interacted with ZnO, forming the PdZn species. This was attributed to the lesser affinity of HEC functional groups to metal ions in comparison with those of PVP.

### 2.2. Catalytic Test

The obtained palladium and palladium–silver catalysts, modified using natural polysaccharides (chitosan (Chit), 2-hydroxyethyl cellulose (HEC) or pectin (Pec)), were tested in the hydrogenation process under mild conditions (0.1 MPa, 40 °C). The hydrogenation of 2-hexyn-1-ol was chosen as a model reaction, with the further prospect of developing catalysts for the hydrogenation of complex acetylene alcohols, the semihydrogenated form of which is used in the production of vitamins, pest pheromones and fragrances [76]. The possible pathways of the reaction are illustrated in Scheme 1. The first-step triple bond of the alkynol was reduced to a C=C double bond. This formed forming *cis/trans*-isomers, which then hydrogenated to alkanol.

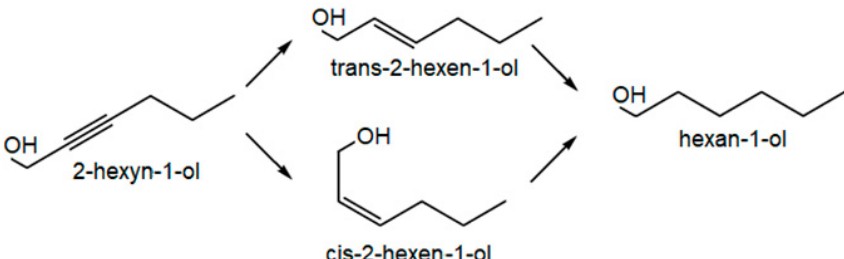

**Scheme 1.** Hydrogenation of 2-hexyn-1-ol.

The test results of the developed catalysts in the hydrogenation process of 2-hexyn-1-ol are given in Table 3. Monometallic palladium catalysts were more active than identical bimetallic Pd-Ag catalysts, but less selective. The reaction rate on bimetallic catalysts decreased depending on the polymer modifier in the following order: HEC > Pec > Chit (Figure 5). The maximum yield of alkenol was observed on the bimetallic PdAg catalyst stabilized with HEC and was 89.4% at a rate conversion of 93.0% (Table 3). The introduction of silver into the HEC-containing catalyst improved selectivity to 2-hexen-1-ol compared with selectivity toward a monometallic Pd catalyst (Figure 6). The *cis*-alkenol selectivity achieved on this catalyst was 97.2% versus 90.6% achieved on Pd-HEC/ZnO.

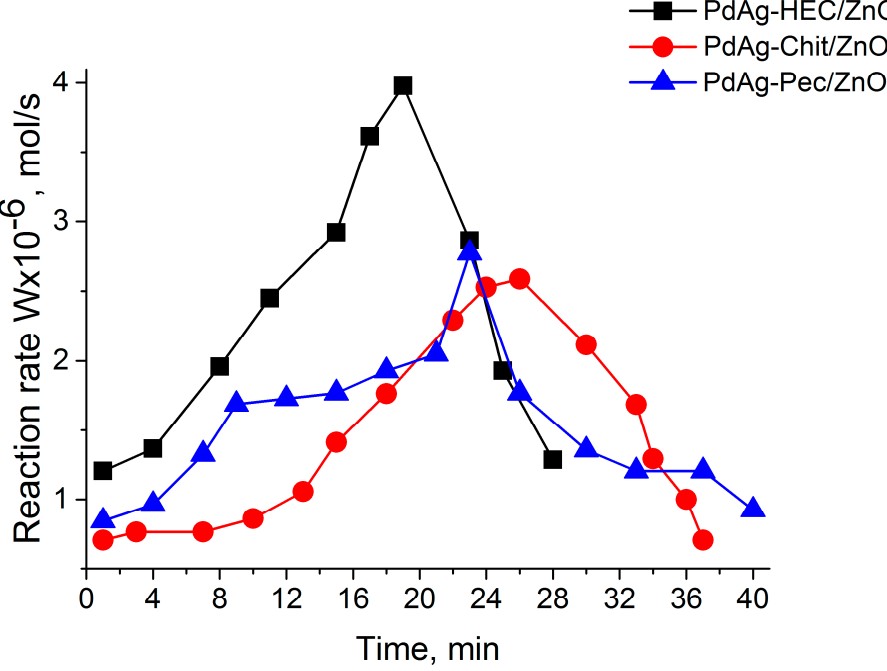

**Figure 5.** The rate of 2-hexyn-1-ol hydrogenation on the PdAg/ZnO catalysts modified with hydroxyethyl cellulose (HEC), chitosan (Chit) and pectin (Pec). Reaction conditions: T = 40 °C, $P_{H2}$ = 1 atm, $m_{cat}$ = 0.05 g, ethanol = 25 mL and $C_{alkynol}$ = 0.09 mol/L.

**Table 3.** The results of 2-hexyn-1-ol hydrogenation on Pd/ZnO and PdAg/ZnO catalysts stabilized with hydroxyethyl cellulose (HEC), chitosan (Chit) and pectin (Pec) *.

| Catalysts | $W_{max} \times 10^{-6}$ (mol s$^{-1}$) | Maximum Yield of *cis*-Hexen-1-ol, % | $S_{cis\text{-hexen-1-ol}}$,% | Conversion, % |
|---|---|---|---|---|
| Pd-HEC/ZnO | 4.3 | 82.8 | 90.6 | 91.4 |
| Pd-Chit/ZnO | 3.1 | 80.1 | 86.3 | 92.8 |
| Pd-Pec/ZnO | 5.4 | 80.9 | 86.1 | 94.0 |
| PdAg-HEC/ZnO | 4.0 | 89.4 | 97.2 | 93.0 |
| PdAg-Chit/ZnO | 2.6 | 85.9 | 92.3 | 93.1 |
| PdAg-Pec/ZnO | 2.8 | 86.4 | 93.5 | 92.4 |

* Reaction conditions presented in Figure 5.

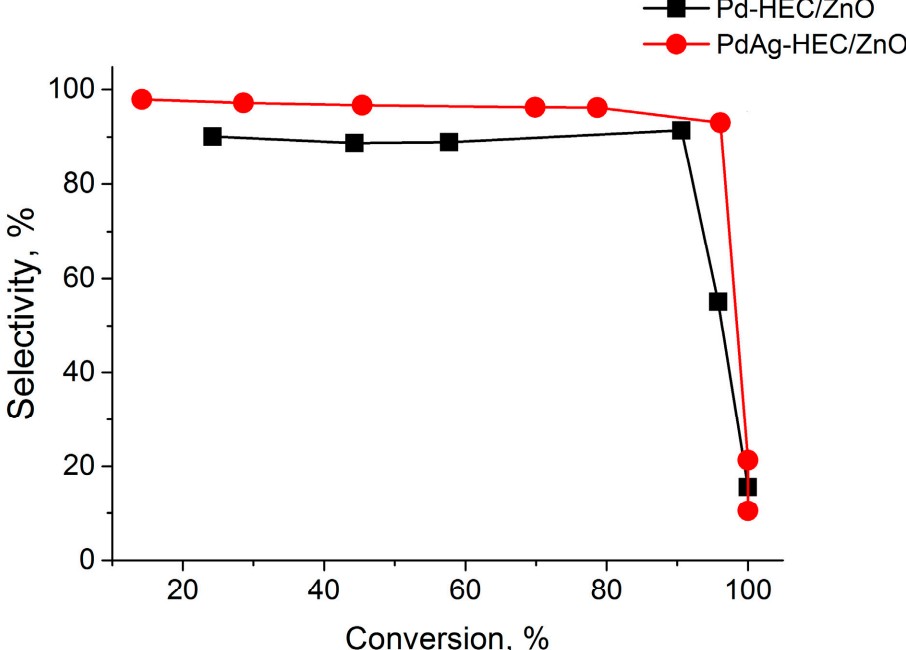

**Figure 6.** Selectivity of 2-hexen-1-ol vs. conversion of 2-hexyn-1-ol on Pd-HEC and PdAg-HEC catalysts supported on ZnO. Reaction conditions are presented in Figure 5.

We determined the optimum temperature and catalyst loading for the PdAg catalysts modified with HEC, Chit and Pec, and the results are shown in Tables 4 and 5. The initial reaction rate increased with the rising temperature up to 40 °C.

Further temperature increases to 50 °C led to a significant decrease in the reaction rate, which could be attributed to the shrinking of the surface polymer layer of the catalyst and the blocking of the active centers [64].

The increase in the catalyst loading from 0.01 g to 0.1 g led to an increase in the reaction rate. The highest rate was achieved on the 0.1 g load of both catalysts. At the same time, the maximum yield and selectivity to *cis*-hexene-1-ol decreased, which was probably due to the accelerated hydrogenation of the *cis*-alkene double bonds into alkanes.

According to the chromatographic analysis, in the presence of HEC-modified catalyst, the conversion of 2-hexyn-1-ol into *cis*-2-hexen-1-ol began in the first half of the process (Figure 7a). After the almost-complete disappearance of acetylenic alcohol in the reaction medium, a process of isomerization of *cis*-alkenol into *trans*-form was initiated and a parallel reduction of double bonds of alkenols to produce saturated alcohol was performed. A similar change in the composition of the reaction mixture was observed when the process was carried out using catalysts containing chitosan. The small difference was that the

isomerization of *cis*- to *trans*-2-hexen-1-ol onto this catalyst started slightly before the complete conversion of the initial alkynol (Figure 7b).

**Table 4.** Effect of variations in temperature on reaction conditions in hydrogenation of 2-hexyn-1-ol *.

| Catalysts | Temperature, °C | $W_{max} \cdot \times 10^{-6}$ (mol s$^{-1}$) | Maximum Yield of *cis*-Hexen-1-ol, % | $S_{cis\text{-hexen-1-ol}}$, % |
|---|---|---|---|---|
| PdAg-Chit/ZnO | 20 | 0.6 | 84.0 | 91.8 |
| | 30 | 2.3 | 76.4 | 93.5 |
| | 40 | 2.6 | 85.9 | 92.3 |
| | 50 | 2.3 | 79.3 | 79.8 |
| PdAg-HEC/ZnO | 20 | 0.4 | 77.0 | 93.0 |
| | 30 | 1.3 | 75.6 | 84.9 |
| | 40 | 4.0 | 89.4 | 97.2 |
| | 50 | 1.6 | 76.2 | 86.7 |
| PdAg-Pec/ZnO | 20 | 0.4 | 78.0 | 91.6 |
| | 30 | 1.9 | 79.6 | 85.6 |
| | 40 | 2.8 | 86.4 | 93.5 |
| | 50 | 1.8 | 78.2 | 87.0 |

* Reaction conditions: T = 20 °C, 30 °C, 40 °C, 50 °C, $P_{H2}$ = 1 atm, $m_{cat}$ = 0.05 g, ethanol = 25 mL and $C_{alkynol}$ = 0.09 mol/L.

**Table 5.** Effect of variations in catalyst loading on reaction conditions in hydrogenation of 2-hexyn-1-ol *.

| Catalysts | Catalyst Loading, g | $W_{max} \cdot \times 10^{-6}$ (mol s$^{-1}$) | Maximum Yield of *cis*-Hexen-1-ol, % | $S_{cis\text{-hexen-1-ol}}$, % |
|---|---|---|---|---|
| **PdAg-Chit/ZnO** | 0.01 | 0.6 | 62.0 | 67.8 |
| | 0.03 | 1.9 | 71.8 | 84.7 |
| | 0.05 | 2.6 | 85.9 | 92.3 |
| | 0.10 | 2.8 | 83.5 | 89.9 |
| **PdAg-HEC/ZnO** | 0.01 | 0.5 | 58.4 | 63.2 |
| | 0.03 | 2.5 | 88.2 | 89.8 |
| | 0.05 | 4.0 | 89.4 | 97.2 |
| | 0.10 | 4.4 | 87.6 | 94.5 |
| **PdAg-Pec/ZnO** | 0.01 | 0.5 | 60.3 | 61.7 |
| | 0.03 | 2.1 | 78.4 | 83.0 |
| | 0.05 | 2.8 | 86.4 | 93.5 |
| | 0.10 | 3.1 | 84.6 | 90.0 |

* Reaction conditions: T = 40 °C, $P_{H2}$ = 1 atm, $m_{cat}$ = 0.01 g, 0.03 g, 0.05 g, 0.10 g, ethanol = 25 mL, $C_{alkynol}$ = 0.09 mol/L.

The reusability of PdAg-Chit/ZnO and PdAg-HEC/ZnO catalysts was studied via the hydrogenation of successive portions of 2-hexyn-1-ol using the same load of catalyst (Figure 8). The high stability was demonstrated by the fact that the reaction rate seen using both catalysts remained nearly the same after at least 10 runs. This could be attributed to the swelling ability of the polymer–metal shell of the catalysts in ethanol and the limitation of the leaching active phase via the prevention of the effect of polymers [64].

Thus, a comparison of the performance of palladium and palladium–silver catalysts, stabilized with the derivatives of natural polysaccharides (Table 5) and pectin, confirmed the potential of their use in the synthesis of metal nanoparticles and their further application in catalysis.

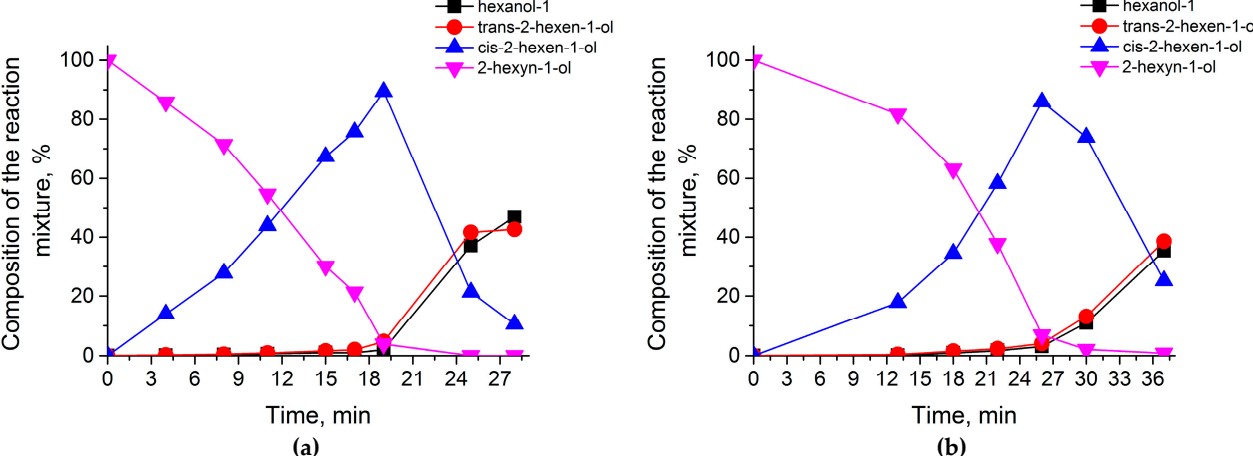

**Figure 7.** Changes in the composition of the reaction mixture during the hydrogenation of 2-hexyn-1-ol in the presence of PdAg-HEC/ZnO (**a**) and PdAg-Chit/ZnO (**b**). Reaction conditions are presented in Figure 5.

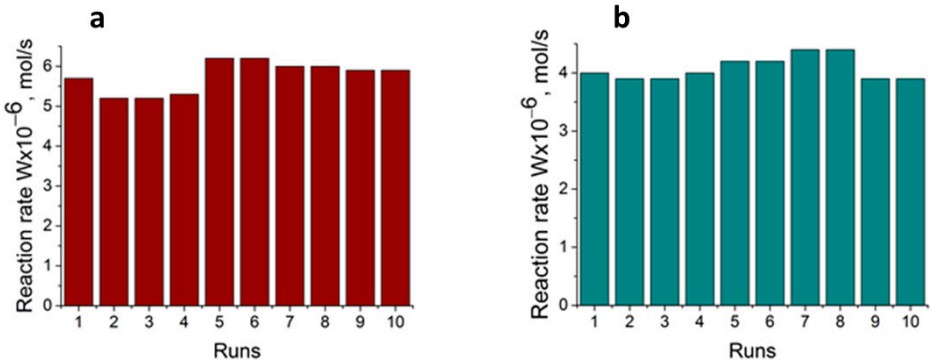

**Figure 8.** Hydrogenation of 10 portions of 2-hexyn-1-ol in presence of PdAg-Chit/ZnO (**a**) and PdAg-HEC/ZnO (**b**). Reaction conditions are presented in Figure 5.

## 3. Materials and Methods

### 3.1. Materials

The chemicals 2-hexyn-1-ol, palladium chloride (PdCl$_2$, 59–60% Pd), silver nitrate (AgNO$_3$, 99.0%), potassium chloride (KCl, reagent grade), 2-hydroxyethyl cellulose (HEC, Mw 90,000), pectin (Pec, Mw 15,000), chitosan (Chit, Mw 250,000) and zinc oxide (chemically pure) were acquired from Sigma-Aldrich, St. Louis, MO, USA. Ethanol (reagent) was purchased from Talgar Alcohol LLP (Talgar, Kazakhstan) and purified via distillation.

### 3.2. Preparation of K$_2$PdCl$_4$ Precursor Solution

A K$_2$PdCl$_4$ precursor solution was prepared by crushing 168.4 mg of palladium (II) chloride and 155.7 mg of KCl in an agate mortar according to the procedure described in [52]. The obtained potassium (II) tetrachloropalladate was dissolved in 50 mL of distilled water. The process was carried out at 70 °C with constant magnetic stirring for 2 h. The concentration of palladium ions in the resulting solution was 0.019 M.

### 3.3. Synthesis of Pd-Polysaccharide/ZnO Catalysts

The nanocatalysts were prepared via an adsorption method in accordance with the procedure described in [64,67]. A 10 mL water solution of $0.9 \times 10^{-2}$ M 2-hydroxyethylcellulose polysaccharide (HEC, Chit or Pec) was added dropwise to the aqueous suspension of inorganic sorbent (1 g ZnO in 15 mL of water) and stirred for 2 h. Then, we added 5 mL of a $0.9 \times 10^{-2}$ M water solution of potassium (II) tetrachloropalladate. This process

was carried out at room temperature with constant stirring for 3 h. The concentration of 2-hydroxyethylcellulose and potassium (II) tetrachloropalladate solutions corresponded to a palladium content of 0.5% and a molar ratio of (Pd:polysaccharide) = 1:1. The synthesized catalysts were kept in the mother liquor for 12–15 h. Then, we conducted washing with distilled water and air-drying. The amount of immobilized palladium was monitored using photoelectrocolorimetry.

### 3.4. Synthesis of PdAg-Polysaccharide/ZnO Catalysts

The method of polysaccharide adsorption onto an inorganic sorbent followed by metal ion deposition was used to prepare 0.5% bimetallic PdAg polysaccharide–inorganic nanocatalysts [64,67]. In total, 5 mL of $0.9 \times 10^{-2}$ M polysaccharide solution (HEC, Chit or Pec) was added to the aqueous suspension of the support (1 g ZnO in 15 mL of water). The preparation process was carried out at room temperature with constant stirring for 2 h. Afterward, palladium salt and then silver salt ($K_2PdCl_4$ and $AgNO_3$) were added under constant stirring conditions. The duration of the process was 3 h. The concentration of potassium (II) tetrachloropalladate and silver (I) nitrate water solutions corresponded to 0.5% metal content (Pd:Ag ratio = 3:1). The amount of polymer for catalyst preparation was calculated at the rate of one transition-metal atom per monomer unit. After keeping the synthesized nanocatalyst in the mother liquor for 12–15 h, it was washed with distilled water and dried in air. The completeness of palladium and silver fixation was monitored via photoelectrocolorimetry and direct potentiometry, respectively.

### 3.5. Characterization of Catalysts

The concentration of metals (Pd, Ag) in nanocatalysts was monitored via the change in the concentration of palladium and silver ions in aqueous solution before and after the immobilization of $Ag^+$ and/or $PdCl_4^{2-}$ on an inorganic support (ZnO) modified using polysaccharide. The quantitative content of Pd in aqueous solutions was detected via photoelectrocolorimetry (PEC). The measurement was carried out using an SF-2000 UV/Vis spectrophotometer (OKB Spektr, St. Petersburg, Russia) according to calibration curves (wavelength $\lambda$ = 425 nm). Ag concentration in aqueous solutions was monitored using the potentiometric method (direct potentiometry method). The measurement was carried out on an ANION 4100 ionometer (Infraspak-Analit, Novosibirsk, Russia) using an ion-selective electrode ELIS-131Ag.

A DRON-4-0.7 powder X-ray diffractometer (Burevestnik, St. Petersburg, Russia), with monochromatized radiation of cobalt Co-K$\alpha$ ($\lambda$ = 0.179 nm), was used to obtain powder X-ray diffractograms.

The specific surface area and pore size distribution of the obtained nanocatalysts were investigated using the low-temperature $N_2$ adsorption–desorption method. The study was carried out on an Accusorb instrument (Micromeritics, Norcross, GA, USA).

The catalyst samples were studied via the FTIR spectroscopic method. A Nicolet iS5 instrument (Thermo Scientific, Waltham, MA, USA) was used to study the samples in the 4000–400 cm$^{-1}$ region via FTIR spectroscopy. A mixture of 1 mg of sample with 100 mg of dry potassium bromide was ground to obtain pellets for IR analysis. The mixture was then pressed into a mold.

Transmission electron microscopy (TEM) micrographs were obtained on a JEM-2100 transmission electron microscope (Jeol, Tokyo, Japan) with an accelerating voltage of 100 kV. Elemental analysis of the nanocatalysts was performed on a JSM-6610LV scanning electron microscope using an EDX detector (Jeol, Tokyo, Japan).

Nanocatalysts were investigated via X-ray photoelectron spectroscopy (XPS) on a Kratos Axis Ultra DLD photoelectron spectrometer (Kratos Analytical LTD, Manchester, UK).

*3.6. Hydrogenation of 2-Hexyn-1-ol*

The hydrogenation process was carried out in a thermostated glass reactor according to the procedure described in Ref [64]. The reaction was carried out in ethanol medium (25 mL) at atmospheric hydrogen pressure and a temperature of 20–50 °C, with intensive stirring (600–700 oscillations per minute). Before hydrogenation, the nanocatalyst (0.05 g) was reduced with hydrogen in the reactor for 30 min under conditions of intensive stirring. After hydrogen treatment, 2.23 mmol (0.09 mol/L) alkynol was added to the reactor. The amount of alkynol corresponded to an uptake of 100 mL of hydrogen. The hydrogenation rate was calculated from hydrogen uptake per unit time. For this purpose, the volume of hydrogen uptake was measured after a certain time interval using a burette connected to the reactor.

To determine the selectivity for the main products of the hydrogenation reaction, syringe samples of the reaction mixture were taken at proper time intervals.

The hydrogenation products were analyzed using gas–liquid chromatography on a Chromos GC1000 chromatograph (Chromos Engineering, Dzerzhinsk, Russia) with a flame ionization detector in isothermal mode. A BP21 capillary column (FFAP) with a polar phase (PEG modified with nitroterephthalate) was used. This device was 50 m in length and 0.32 mm in inner diameter. The column temperature was 90 °C, the injector temperature was 200 °C and helium served as the carrier gas. A total of 0.2 mL of sample was investigated. Selectivity for alkenol was calculated as the fraction of the target product present in the reaction products at a given degree of substrate conversion.

To determine the stability of the catalysts, hydrogenation of successive portions of alkynol (2.23–4.46 mmol) was carried out for the same nanocatalyst sample (0.05 g).

**4. Conclusions**

Pd/ZnO and PdAg/ZnO nanocatalysts, modified with pectin (Pec), 2-hydroxyethyl cellulose (HEC) and chitosan (Chit), were prepared using a green one-pot method via the introduction of water solutions of polysaccharide and metal salts into a water suspension of ZnO under ambient conditions. A study of the resulting catalysts using spectrophotometry, potentiometry, elemental analysis, XRD, IRS, BET, TEM and XPS methods indicated both that polysaccharides and metal (Pd and Ag) ions were quantitatively adsorbed onto zinc oxide and the polymers interacted with both ZnO and the active-phase particles formed. In the case of Pd-HEC/ZnO, the interaction of a small amount of Pd with ZnO and the formation of PdZn species was also observed. This suggests that, by varying the polymer nature in such a variety of catalysts, it is possible to regulate the composition of active-phase particles.

The catalysts showed excellent activity in the hydrogenation of model 2-hexyn-1-ol substrates at 40 °C and 1 atm of $H_2$. In this comparative study, the maximum selectivity for 2-hexen-1-ol (97.2%) was obtained in the presence of a bimetallic PdAg-HEC/ZnO catalyst. Monometallic Pd catalysts showed high activity, but lower selectivity, than Pd-Ag catalysts.

The hydrogenation reaction takes place in a swollen bulk metal–polymer surface layer, increasing the lifetime of PdAg-HEC/ZnO. Thus, the catalyst combines the advantages of both homogeneous and heterogeneous catalysts.

The excellent performance at low catalyst loadings and mild reaction conditions makes polysaccharides containing metal nanocatalysts highly attractive for further improvement. Additional testing should be performed in both the hydrogenation of different types of unsaturated organic compounds and other important catalytic processes.

**Supplementary Materials:** The following supporting information can be downloaded at: https://www.mdpi.com/article/10.3390/catal13111403/s1, Table S1: Results of adsorption of $Pd^{2+}$ and $Ag^+$ ions on polymer-modified ZnO; Table S2: Analysis of electron diffraction pattern from TEM of 0.5%PdAg-HEC/ZnO catalyst [77]; Table S3: Analysis of electron diffraction pattern from TEM of 0.5%Pd-HEC/ZnO catalyst [77].

**Author Contributions:** Conceptualization, A.S.A. and A.K.Z.; methodology, E.T.T.; software, F.U.B. and A.I.J.; validation and formal analysis, A.K.Z., E.T.T. and A.S.A.; investigation, A.I.J. and F.U.B.; resources, A.K.Z., E.T.T. and A.S.A.; data curation, F.U.B.; writing—original draft preparation, A.K.Z., E.T.T. and A.S.A.; writing—review and editing, A.K.Z. and A.S.A.; visualization, A.I.J. and F.U.B.; supervision, project administration and funding acquisition, A.K.Z. All authors have read and agreed to the published version of the manuscript.

**Funding:** This research was funded by the Committee of Science of the Ministry of Science and Higher Education of the Republic of Kazakhstan (grant no.: AP09259638).

**Data Availability Statement:** The data that support the findings of this study are available from the corresponding author upon reasonable request.

**Conflicts of Interest:** The authors declare no conflict of interest.

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
