# Peer review of "Polysaccharide-Stabilized PdAg Nanocatalysts for Hydrogenation of 2-Hexyn-1-ol"

_catalysts, doi:10.3390/catal13111403_

Round 1

Reviewer 1 Report

Comments and Suggestions for Authors

This manuscript reports a new one-pot green technique was used for preparation of polysaccharide-based Pd- and PdAg nanocatalysts and the catalyst performed well for liquid-phase hydrogenation of 2-hexyn-1-ol. This is an interesting and noteworthy work. However, some of the authors' ideas lack relevant representations to support them. I think this paper needs to be revised before considered to be accepted. The detailed comments are listed below:

1.     In the manuscript, on page 4, there is a lack of literature support regarding the attribution of signal peaks in infrared spectroscopy, and it is recommended to cite relevant literature.

2.     In the manuscript, the text of the figure note for figure 2 is incorrectly formatted, and "ABSORBANCE" and "Wavenumbers (cm-1)" in the figure should be changed to "Absorbance (a. u.)" and "Wavenumbers (cm-1)", respectively.

3.     In the manuscript, on pages 4 and 5, in the TEM test results about the catalyst Pd-HCE/ZnO, the Mapping test results are not provided, which cannot sufficiently prove that the surface of the catalyst is Pd nanoparticles, please supplement the relevant experiments.

4.     The manuscript lacks studies on the reaction sites on the catalysts as well as intermediate species during hydrogena-tion of 2-hexyn-1-ol reactions with PdAg-HEC/ZnO and PdAg-Chit/ZnO catalysts. It is suggested that the reaction pathways can be explored by characterisation such as NMR.

5.     In the manuscript, on page 6, the authors argue that "that binding energy of Pd0 has a positive shift (ca. 0.6 eV) ...... formation of PdZn species" only by the results of the XPS test, which lacks relevant characterisation support. It is suggested that the authors could characterise it by HRTEM.

Author Response

Responses to the reviewer's comments in the attached file.

Reviewer 2 Report

Comments and Suggestions for Authors

The paper by Zharmagambetova et al. provided the data on catalytic activity of Pd containing catalysts stabilized by different polysaccharides. Before the manuscript can be accepted to publication it must be vastly revised.

1. A serious concern is raised by the absence of structural characterization data on most of the catalysts synthesized. The authors provided a full characterization on Pd-HEC/ZnO catalyst, however, the Pd-Ag-based catalysts were found to be the most active. Meanwhile TEM images, XPS spectra for those composites are missed. Similarly, no information on the influence of different polysaccharide support on the structural properties of the catalysts is given. Does the support influence the size and morphology of Pd NPs, Pd2+/Pd0 ratio? These parameters may affect the catalytic activity significantly. The data presented include the HEC-based composites only, while the structural properties of pectin- and chitosan-based catalysts are not discussed.

2. The IR spectra presented in Figure 2 are very confusing. In particular, the different width and saturation of the line in a single spectrum, a very low resolution. I recommend to provide a figure with a necessary quality and move it to Supporting Information.

3. Table 1 is not informative, it should be moved to Supporting Information.

4. The Zn2+ binding energies in XPS spectrum (Figure 4) differ significantly from the literature data. The typical binding energies of Zn2+ in ZnO are 1022.5 and 1045.9 eV for 2p3/2 and 2p1/2, respectively (Pauly, N., et al., XPS primary excitation spectra of Zn 2p, Fe 2p, and Ce 3d from ZnO, α-Fe2O3, and CeO2. Surface and Interface Analysis, 2019. 51(3): p. 353-360; J.F. Moulder, W.F. Stickle, P.E. Sobol, K.D. Bomben, Handbook of X-ray Photoelectron Spectroscopy), while binding energy of 1020.4 eV for 2p3/2 is observed in the work. The spin-orbit splitting energy (Zn 2p3/2 – Zn 2p1/2  = 24.6 eV) does not also coincide with that of ZnO (23.1 eV). The position of the 2p1/2 peak at the Figure 4 does not match the 1045 eV as indicated at the picture.

5. The presentation of data in Table 5 should be modified. The present form is awkward and makes reading  the Table difficult.

6. Data on the catalytic activity of the pectin-based catalysts should be added in the Table 5.

7. The catalytic activity of the composites synthesized in the manuscript should be compared with the other Pd-containing polysaccharide-based systems.

8. The introduction should be revised. Some crucial information describing the state of the art in the field of polysaccharide-based catalysts should be disclosed. In particular, the catalytic activity of polysaccharide-based catalysts developed earlier, their major advantages and shortcomings, the novelty of the catalysts synthesized in the manuscript over the previously reported polysaccharide-based  systems, the importance of hydrogenation of 2-hexyn-1-ol should be discussed.

9. The manuscript should be thoroughly checked to exclude the grammatical and spelling errors. Some grammatical errors found (but not limited to) in the manuscript upon the brief checking: bigan (p.8), stabilised (p.9), natutal (p.9).

Comments on the Quality of English Language

English should be thoroughly checked to exclude the grammatical and spelling errors.

Author Response

(The authors gave the same response as above.)

Round 2

Reviewer 1 Report

Comments and Suggestions for Authors

After authors revised, the manuscript has largely met the requirements of the journal. However, there is one area that needs to be revised, as described below:

1.     In Figure 3a, the Pd nanoparticles could be circled in the TEM about Pd-HCE/ZnO, please make them clearly distinguishable.

Author Response

Thank you very much for taking the time to review and for the opportunity to revise our manuscript entitled “Polysaccharide-Stabilized PdAg Nanocatalysts for Hydrogenation of 2-Hexyn-1-ol”. A response to the comment attached below.
